# Adversarial Training for Free!

**Ali Shafahi**
University of Maryland
ashafahi@cs.umd.edu

**Mahyar Najibi**
University of Maryland
najibi@cs.umd.edu

**Amin Ghiasi**
University of Maryland
amin@cs.umd.edu

**Zheng Xu**
University of Maryland
xuzh@cs.umd.edu

**John Dickerson**
University of Maryland
john@cs.umd.edu

**Christoph Studer**
Cornell University
studer@cornell.edu

**Larry S. Davis**
University of Maryland
lsd@umiacs.umd.edu

**Gavin Taylor**
United States Naval Academy
taylor@usna.edu

**Tom Goldstein**
University of Maryland
tomg@cs.umd.edu

## Abstract

*Adversarial training*, in which a network is trained on adversarial examples, is one of the few defenses against adversarial attacks that withstands strong attacks. Unfortunately, the high cost of generating strong adversarial examples makes standard adversarial training impractical on large-scale problems like ImageNet. We present an algorithm that eliminates the overhead cost of generating adversarial examples by recycling the gradient information computed when updating model parameters. Our "free" adversarial training algorithm achieves comparable robustness to PGD adversarial training on the CIFAR-10 and CIFAR-100 datasets at negligible additional cost compared to natural training, and can be 7 to 30 times faster than other strong adversarial training methods. Using a single workstation with 4 P100 GPUs and 2 days of runtime, we can train a robust model for the large-scale ImageNet classification task that maintains 40% accuracy against PGD attacks.

## 1   Introduction

Deep learning has been widely applied to various computer vision tasks with excellent performance. Prior to the realization of the adversarial example phenomenon by Biggio et al. [2013], Szegedy et al. [2013], model performance on clean examples was the the main evaluation criteria. However, in security-critical applications, robustness to adversarial attacks has emerged as a critical factor.

A robust classifier is one that correctly labels adversarially perturbed images. Alternatively, robustness may be achieved by detecting and rejecting adversarial examples [Ma et al., 2018, Meng and Chen, 2017, Xu et al., 2017]. Recently, Athalye et al. [2018] broke a complete suite of allegedly robust defenses, leaving *adversarial training*, in which the defender augments each minibatch of training data with adversarial examples [Madry et al., 2017], among the few that remain resistant to attacks. Adversarial training is time-consuming—in addition to the gradient computation needed to update the network parameters, each stochastic gradient descent (SGD) iteration requires multiple gradient computations to produce adversarial images. In fact, it takes 3-30 times longer to form a robust network with adversarial training than forming a non-robust equivalent. Put simply, the actual slowdown factor depends on the number of gradient steps used for adversarial example generation.

The high cost of adversarial training has motivated a number of alternatives. Some recent works replace the perturbation generation in adversarial training with a parameterized generator network

[Baluja and Fischer, 2018, Poursaeed et al., 2018, Xiao et al., 2018]. This approach is slower than standard training, and problematic on complex datasets, such as ImageNet, for which it is hard to produce highly expressive GANs that cover the entire image space. Another popular defense strategy is to regularize the training loss using label smoothing, logit squeezing, or a Jacobian regularization [Shafahi et al., 2019a, Mosbach et al., 2018, Ross and Doshi-Velez, 2018, Hein and Andriushchenko, 2017, Jakubovitz and Giryes, 2018, Yu et al., 2018]. These methods have not been applied to large-scale problems, such as ImageNet, and can be applied in parallel to adversarial training.

Recently, there has been a surge of certified defenses [Wong and Kolter, 2017, Wong et al., 2018, Raghunathan et al., 2018a,b, Wang et al., 2018]. These methods were mostly demonstrated for small networks, low-res datasets, and relatively small perturbation budgets ($\epsilon$). Lecuyer et al. [2018] propose randomized smoothing as a certified defense and which was later improved by Li et al. [2018a]. Cohen et al. [2019] prove a tight robustness guarantee under the $\ell_2$ norm for smoothing with Gaussian noise. Their study was the first certifiable defense for the ImageNet dataset [Deng et al., 2009]. They claim to achieve 12% robustness against non-targeted attacks that are within an $\ell_2$ radius of 3 (for images with pixels in $[0, 1]$). This is roughly equivalent to an $\ell_\infty$ radius of $\epsilon = 2$ when pixels lie in $[0, 255]$.

Adversarial training remains among the most trusted defenses, but it is nearly intractable on large-scale problems. Adversarial training on high-resolution datasets, including ImageNet, has only been within reach for research labs having hundreds of GPUs[1]. Even on reasonably-sized datasets, such as CIFAR-10 and CIFAR-100, adversarial training is time consuming and can take multiple days.

**Contributions**

We propose a fast adversarial training algorithm that produces robust models with almost no extra cost relative to natural training. The key idea is to update both the model parameters and image perturbations using one simultaneous backward pass, rather than using separate gradient computations for each update step. Our proposed method has the same computational cost as conventional natural training, and can be 3-30 times faster than previous adversarial training methods [Madry et al., 2017, Xie et al., 2019]. Our robust models trained on CIFAR-10 and CIFAR-100 achieve accuracies comparable and even slightly exceeding models trained with conventional adversarial training when defending against strong PGD attacks.

We can apply our algorithm to the large-scale ImageNet classification task on a single workstation with four P100 GPUs in about two days, achieving 40% accuracy against non-targeted PGD attacks. To the best of our knowledge, our method is the first to successfully train a robust model for ImageNet based on the non-targeted formulation and achieves results competitive with previous (significantly more complex) methods [Kannan et al., 2018, Xie et al., 2019].

## 2 Non-targeted adversarial examples

Adversarial examples come in two flavors: *non-targeted* and *targeted*. Given a fixed classifier with parameters $\theta$, an image $x$ with true label $y$, and classification proxy loss $l$, a bounded *non-targeted* attack sneaks an example out of its natural class and into another. This is done by solving

$$\max_\delta \quad l(x + \delta, y, \theta), \quad \text{subject to} \ \ ||\delta||_p \leq \epsilon, \tag{1}$$

where $\delta$ is the adversarial perturbation, $||.||_p$ is some $\ell_p$-norm distance metric, and $\epsilon$ is the adversarial manipulation budget. In contrast to non-targeted attacks, a *targeted* attack scooches an image into a specific class of the attacker's choice.

In what follows, we will use non-targeted adversarial examples both for evaluating the robustness of our models and also for adversarial training. We briefly review some of the closely related methods for generating adversarial examples. In the context of $\ell_\infty$-bounded attacks, the Fast Gradient Sign Method (FGSM) by Goodfellow et al. [2015] is one of the most popular non-targeted methods that uses the sign of the gradients to construct an adversarial example in one iteration:

$$x_{adv} = x + \epsilon \cdot sign(\nabla_x l(x, y, \theta)). \tag{2}$$

The Basic Iterative Method (BIM) by Kurakin et al. [2016a] is an iterative version of FGSM. The PGD attack is a variant of BIM with uniform random noise as initialization, which is recognized by Athalye et al. [2018] to be one of the most powerful first-order attacks. The initial random noise was first studied by Tramèr et al. [2017] to enable FGSM to attack models that rely on "gradient masking." In the PGD attack algorithm, the number of iterations $K$ plays an important role in the strength of attacks, and also the computation time for generating adversarial examples. In each iteration, a complete forward and backward pass is needed to compute the gradient of the loss with respect to the image. Throughout this paper we will refer to a $K$-step PGD attack as PGD-$K$.

# 3    Adversarial training

Adversarial training can be traced back to [Goodfellow et al., 2015], in which models were hardened by producing adversarial examples and injecting them into training data. The robustness achieved by adversarial training depends on the strength of the adversarial examples used. Training on fast non-iterative attacks such as FGSM and Rand+FGSM only results in robustness against non-iterative attacks, and not against PGD attacks [Kurakin et al., 2016b, Madry et al., 2017]. Consequently, Madry et al. [2017] propose training on multi-step PGD adversaries, achieving state-of-the-art robustness levels against $\ell_\infty$ attacks on MNIST and CIFAR-10 datasets.

While many defenses were broken by Athalye et al. [2018], PGD-based adversarial training was among the few that withstood strong attacks. Many other defenses build on PGD adversarial training or leverage PGD adversarial generation during training. Examples include Adversarial Logit Pairing (ALP) [Kannan et al., 2018], Feature Denoising [Xie et al., 2019], Defensive Quantization [Lin et al., 2019], Thermometer Encoding [Buckman et al., 2018], PixelDefend [Song et al., 2017], Robust Manifold Defense [Ilyas et al., 2017], L2-nonexpansive nets [Qian and Wegman, 2018], Jacobian Regularization [Jakubovitz and Giryes, 2018], Universal Perturbation [Shafahi et al., 2018], and Stochastic Activation Pruning [Dhillon et al., 2018].

We focus on the min-max formulation of adversarial training [Madry et al., 2017], which has been theoretically and empirically justified. This widely used $K$-PGD adversarial training algorithm has an inner loop that constructs adversarial examples by PGD-$K$, while the outer loop updates the model using minibatch SGD on the generated examples. In the inner loop, the gradient $\nabla_x l(x_{adv}, y, \theta)$ for updating adversarial examples requires a forward-backward pass of the entire network, which has similar computation cost as calculating the gradient $\nabla_\theta l(x_{adv}, y, \theta)$ for updating network parameters. Compared to natural training, which only requires $\nabla_\theta l(x, y, \theta)$ and does not have an inner loop, K-PGD adversarial training needs roughly $K + 1$ times more computation.

# 4    "Free" adversarial training

$K$-PGD adversarial training [Madry et al., 2017] is generally slow. For example, the 7-PGD training of a WideResNet [Zagoruyko and Komodakis, 2016] on CIFAR-10 in Madry et al. [2017] takes about four days on a Titan X GPU. To scale the algorithm to ImageNet, Xie et al. [2019] and Kannan et al. [2018] had to deploy large GPU clusters at a scale far beyond the reach of most organizations.

Here, we propose *free adversarial training*, which has a negligible complexity overhead compared to natural training. Our free adversarial training algorithm (alg. 1) computes the ascent step by re-using the backward pass needed for the descent step. To update the network parameters, the current training minibatch is passed forward through the network. Then, the gradient with respect to the network parameters is computed on the backward pass. When the "free" method is used, the gradient of the loss with respect to the input image is also computed on this same backward pass.

Unfortunately, this approach does not allow for multiple adversarial updates to be made to the same image without performing multiple backward passes. To overcome this restriction, we propose a minor yet nontrivial modification to training: train on the same minibatch $m$ times in a row. Note that we divide the number of epochs by $m$ such that the overall number of training iterations remains constant. This strategy provides multiple adversarial updates to each training image, thus providing strong/iterative adversarial examples. Finally, when a new minibatch is formed, the perturbation generated on the previous minibatch is used to warm-start the perturbation for the new minibatch.

**Algorithm 1** "Free" Adversarial Training (Free-$m$)

---

**Require:** Training samples $X$, perturbation bound $\epsilon$, learning rate $\tau$, hop steps $m$
1: Initialize $\theta$
2: $\delta \leftarrow 0$
3: **for** epoch $= 1 \ldots N_{ep}/m$ **do**
4:     **for** minibatch $B \subset X$ **do**
5:         **for** i $= 1 \ldots m$ **do**
6:             Update $\theta$ with stochastic gradient descent
7:                 $g_\theta \leftarrow \mathbb{E}_{(x,y)\in B}[\nabla_\theta \, l(x+\delta, y, \theta)]$
8:                 $g_{adv} \leftarrow \nabla_x \, l(x+\delta, y, \theta)]$
9:                 $\theta \leftarrow \theta - \tau g_\theta$
10:           Use gradients calculated for the minimization step to update $\delta$
11:              $\delta \leftarrow \delta + \epsilon \cdot \mathrm{sign}(g_{adv})$
12:              $\delta \leftarrow \mathrm{clip}(\delta, -\epsilon, \epsilon)$
13:         **end for**
14:     **end for**
15: **end for**

---

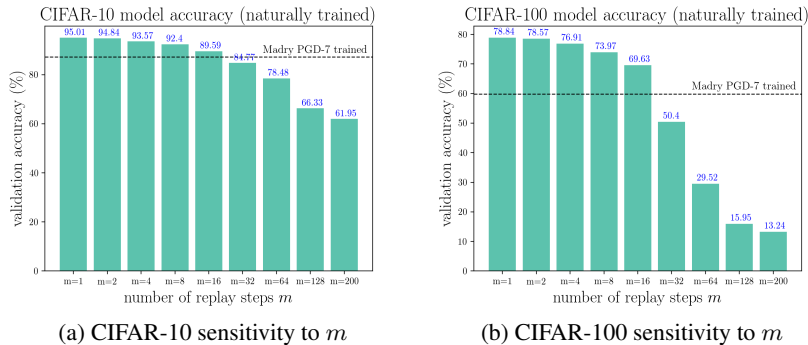

(a) CIFAR-10 sensitivity to $m$          (b) CIFAR-100 sensitivity to $m$

Figure 1: Natural validation accuracy of Wide Resnet 32-10 models using varied mini-batch replay parameters $m$. Here $m = 1$ corresponds to natural training. For large $m$'s, validation accuracy drops drastically. However, small $m$'s have little effect. For reference, CIFAR-10 and CIFAR-100 models that are 7-PGD adversarially trained have natural accuracies of 87.25% and 59.87%, respectively.

**The effect of mini-batch replay on natural training**

While the hope for alg. 1 is to build robust models, we still want models to perform well on natural examples. As we increase $m$ in alg. 1, there is risk of increasing generalization error. Furthermore, it may be possible that catastrophic forgetting happens. Consider the worst case where all the "informative" images of one class are in the first few mini-batches. In this extreme case, we do not see useful examples for most of the epoch, and forgetting may occur. Consequently, a natural question is: how much does mini-batch replay hurt generalization?

To answer this question, we *naturally* train wide-resnet 32-10 models on CIFAR-10 and CIFAR-100 using different levels of replay. Fig. 1 plots clean validation accuracy as a function of the replay parameter $m$. We see some dropoff in accuracy for small values of $m$. Note that a small compromise in accuracy is acceptable given a large increase in robustness due to the fundamental tradeoffs between robustness and generalization [Tsipras et al., 2018, Zhang et al., 2019a, Shafahi et al., 2019b]. As a reference, CIFAR-10 and CIFAR-100 models that are 7-PGD adversarially trained have natural accuracies of 87.25% and 59.87%, respectively. These same accuracies are exceeded by natural training with $m = 16$. We see in section 5 that good robustness can be achieved using "free" adversarial training with just $m \leq 10$.

Table 1: Validation accuracy and robustness of CIFAR-10 models trained with various methods.

| Training | Evaluated Against | | | | | Train Time (min) |
|---|---|---|---|---|---|---|
| | Nat. Images | PGD-20 | PGD-100 | CW-100 | 10 restart PGD-20 | |
| Natural | **95.01%** | 0.00% | 0.00% | 0.00% | 0.00% | **780** |
| Free $m = 2$ | 91.45% | 33.92% | 33.20% | 34.57% | 33.41% | 816 |
| Free $m = 4$ | 87.83% | 41.15% | 40.35% | 41.96% | 40.73% | 800 |
| Free $m = 8$ | 85.96% | **46.82%** | **46.19%** | **46.60%** | **46.33%** | 785 |
| Free $m = 10$ | 83.94% | 46.31% | 45.79% | 45.86% | 45.94% | 785 |
| 7-PGD trained | 87.25% | 45.84% | 45.29% | 46.52% | 45.53% | 5418 |

Table 2: Validation accuracy and robustness of CIFAR-100 models trained with various methods.

| Training | Evaluated Against | | | Training Time (minutes) |
|---|---|---|---|---|
| | Natural Images | PGD-20 | PGD-100 | |
| Natural | **78.84%** | 0.00% | 0.00% | 811 |
| Free $m = 2$ | 69.20% | 15.37% | 14.86% | 816 |
| Free $m = 4$ | 65.28% | 20.64% | 20.15% | **767** |
| Free $m = 6$ | 64.87% | 23.68% | 23.18% | 791 |
| Free $m = 8$ | 62.13% | **25.88%** | **25.58%** | 780 |
| Free $m = 10$ | 59.27% | 25.15% | 24.88% | 776 |
| Madry *et al.* (2-PGD trained) | 67.94% | 17.08% | 16.50% | 2053 |
| Madry *et al.* (7-PGD trained) | 59.87% | 22.76% | 22.52% | 5157 |

# 5 Robust models on CIFAR-10 and 100

In this section, we train robust models on CIFAR-10 and CIFAR-100 using our "free" adversarial training ( alg. 1) and compare them to K-PGD adversarial training [2,3]. We find that free training is able to achieve state-of-the-art robustness on the CIFARs without the overhead of standard PGD training.

**CIFAR-10**

We train various CIFAR-10 models using the Wide-Resnet 32-10 model and standard hyper-parameters used by Madry et al. [2017]. In the proposed method (alg. 1), we repeat (*i.e.* replay) each minibatch $m$ times before switching to the next minibatch. We present the experimental results for various choices of $m$ in table 1. Training each of these models costs roughly the same as natural training since we preserve the same number of iterations. We compare with the 7-PGD adversarially trained model from Madry et al. [2017] [4], whose training requires roughly 7× more time than all of our free training variations. We attack all models using PGD attacks with $K$ iterations on both the cross-entropy loss (PGD-$K$) and the Carlini-Wagner loss (CW-$K$) [Carlini and Wagner, 2017]. We test using the PGD-20 attack following Madry et al. [2017], and also increase the number of attack iterations and employ random restarts to verify robustness under stronger attacks. To measure the sensitivity of our method to initialization, we perform five trials for the Free-$m = 8$ case and find that our results are insensitive. The natural accuracy is 85.95±0.14 and robustness against a 20-random restart PGD-20 attack is 46.49±0.19. Note that gradient free-attacks such as SPSA will result in inferior results for adversarially trained models in comparison to optimization based attacks such as PGD as noted by Uesato et al. [2018]. Gradient-free attacks are superior in settings where the defense works by masking or obfuscating the gradients.

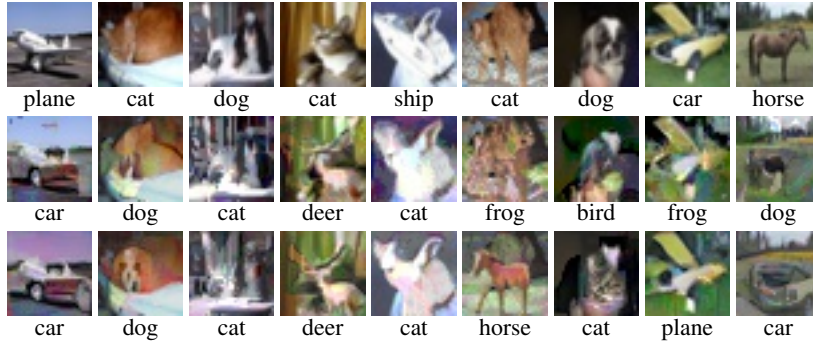

| plane | cat | dog | cat | ship | cat | dog | car | horse |

| car | dog | cat | deer | cat | frog | bird | frog | dog |

| car | dog | cat | deer | cat | horse | cat | plane | car |

Figure 2: Attack images built for adversarially trained models look like the class into which they get misclassified. We display the last 9 CIFAR-10 clean validation images (top row) and their adversarial examples built for a 7-PGD adversarially trained (middle) and our "free" trained (bottom) models.

Our "free training" algorithm successfully reaches robustness levels comparable to a 7-PGD adversarially trained model. As we increase $m$, the robustness is increased at the cost of validation accuracy on natural images. Additionally note that we achieve reasonable robustness over a wide range of choices of the main hyper-parameter of our model, $10 \geq m > 2$, and the proposed method is significantly faster than 7-PGD adversarial training. Recently, a new method called YOPO [Zhang et al., 2019b] has been proposed for speeding up adversarial training, in their CIFAR-10 results they use a wider networks (WRN-34-10) with larger batch-sizes (256). As shown in our supplementary, both of these factors increase robustness. To do a direct comparison, we a train WRN-34-10 using $m = 10$ and batch-size=256. We match their best reported result (48.03% against PGD-20 attacks for "Free" training *v.s.* 47.98% for YOPO 5-3).

**CIFAR-100**

We also study the robustness results of "free training" on CIFAR-100 which is a more difficult dataset with more classes. As we will see in sec. 4, training with large $m$ values on this dataset hurts the natural validation accuracy more in comparison to CIFAR-10. This dataset is less studied in the adversarial machine learning community and therefore for comparison purposes, we adversarially train our own Wide ResNet 32-10 models for CIFAR-100. We train two robust models by varying $K$ in the $K$-PGD adversarial training algorithm. One is trained on PGD-2 with a computational cost almost $3\times$ that of free training, and the other is trained on PGD-7 with a computation time roughly $7\times$ that of free training. We adopt the code for adversarial training from Madry et al. [2017], which produces state-of-the-art robust models on CIFAR-10. We summarize the results in table. 2.

We see that "free training" exceeds the accuracy on both natural images and adversarial images when compared to traditional adversarial training. Similar to the effect of increasing $m$, increasing $K$ in $K$-PGD adversarial training results in increased robustness at the cost of clean validation accuracy. However, unlike the proposed "free training" where increasing $m$ has no extra cost, increasing $K$ for standard $K$-PGD substantially increases training time.

## 6  Does "free" training behave like standard adversarial training?

Here, we analyze two properties that are associated with PGD adversarially trained models: The interpretability of their gradients and the flatness of their loss surface. We find that "free" training enjoys these benefits as well.

**Generative behavior for largely perturbed examples**

Tsipras et al. [2018] observed that hardened classifiers have interpretable gradients; adversarial examples built for PGD trained models often look like the class into which they get misclassified. Fig. 2 plots "weakly bounded" adversarial examples for the CIFAR-10 7-PGD adversarially trained model [Madry et al., 2017] and our free $m = 8$ trained model. Both models were trained to resist $\ell_\infty$ attacks with $\epsilon = 8$. The examples are made using a 50 iteration BIM attack with $\epsilon = 30$ and $\epsilon_s = 2$.

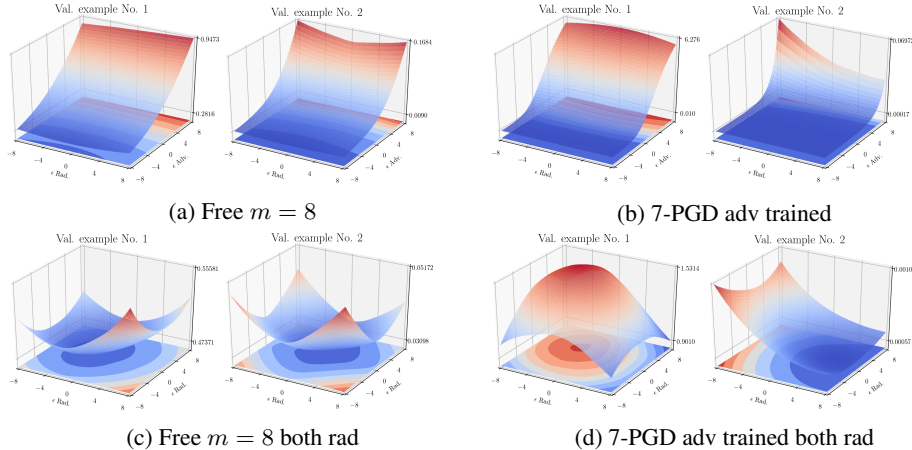

(a) Free $m = 8$  (b) 7-PGD adv trained

(c) Free $m = 8$ both rad  (d) 7-PGD adv trained both rad

Figure 3: The loss surface of a 7-PGD adversarially trained model and our "free" trained model for CIFAR-10 on the first 2 validation images. In (a) and (b) we display the cross-entropy loss projected on one random (Rademacher) and one adversarial direction. In (c) and (d) we display the the cross entropy loss projected along two random directions. Both training methods behave similarly and do not operate by masking the gradients as the adversarial direction is indeed the direction where the cross-entropy loss changes the most.

"Free training" maintains generative properties, as our model's adversarial examples resemble the target class.

**Smooth and flattened loss surface**

Another property of PGD adversarial training is that it flattens and smoothens the loss landscape. In contrast, some defenses work by "masking" the gradients, i.e., making it difficult to identify adversarial examples using gradient methods, even though adversarial examples remain present. Reference Engstrom et al. [2018] argues that gradient masking adds little security. We show in fig. 3a that free training does not operate by masking gradients using a rough loss surface. In fig. 3 we plot the cross-entropy loss projected along two directions in image space for the first few validation examples of CIFAR-10 [Li et al., 2018b]. In addition to the loss of the free $m = 8$ model, we plot the loss of the 7-PGD adversarially trained model for comparison.

# 7  Robust ImageNet classifiers

ImageNet is a large image classification dataset of over 1 million high-res images and 1000 classes (Russakovsky et al. [2015]). Due to the high computational cost of ImageNet training, only a few research teams have been able to afford building robust models for this problem. Kurakin et al. [2016b] first hardened ImageNet classifiers by adversarial training with non-iterative attacks.[5] Adversarial training was done using a targeted FGSM attack. They found that while their model became robust against targeted non-iterative attacks, the targeted BIM attack completely broke it.

Later, Kannan et al. [2018] attempted to train a robust model that withstands targeted PGD attacks. They trained against 10 step PGD targeted attacks (a process that costs 11 times more than natural training) to build a benchmark model. They also generated PGD targeted attacks to train their adversarial logit paired (ALP) ImageNet model. Their baseline achieves a top-1 accuracy of $3.1\%$ against PGD-20 targeted attacks with $\epsilon = 16$. Very recently, Xie et al. [2019] trained a robust ImageNet model against targeted PGD-30 attacks, with a cost $31\times$ that of natural training. Training this model required a distributed implementation on 128 GPUs with batch size 4096. Their robust ResNet-101 model achieves a top-1 accuracy of $35.8\%$ on targeted PGD attacks with many iterations.

Table 3: ImageNet validation accuracy and robustness of ResNet-50 models trained with various replay parameters and $\epsilon = 2$.

| Training | Evaluated Against | | | |
|---|---|---|---|---|
| | Natural Images | PGD-10 | PGD-50 | PGD-100 |
| Natural | **76.038%** | 0.166% | 0.052% | 0.036% |
| Free $m=2$ | 71.210% | 37.012% | 36.340% | 36.250% |
| Free $m=4$ | 64.446% | **43.522%** | **43.392%** | **43.404%** |
| Free $m=6$ | 60.642% | 41.996% | 41.900% | 41.892% |
| Free $m=8$ | 58.116% | 40.044% | 40.008% | 39.996% |

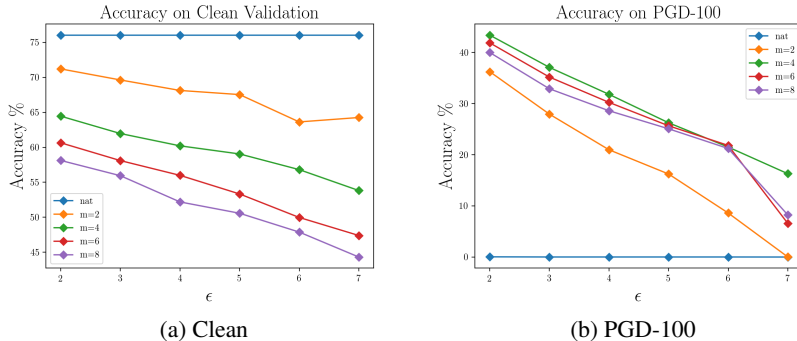

(a) Clean            (b) PGD-100

Figure 4: The effect of the perturbation bound $\epsilon$ and the mini-batch replay hyper-parameter $m$ on the robustness achieved by free training.

**Free training results**

Our alg. 1 is designed for non-targeted adversarial training. As Athalye et al. [2018] state, defending on this task is important and more challenging than defending against targeted attacks, and for this reason smaller $\epsilon$ values are typically used. Even for $\epsilon = 2$ (the smallest $\epsilon$ we consider defensing against), a PGD-50 non-targeted attack on a natural model achieves roughly $0.05\%$ top-1 accuracy. To put things further in perspective, Uesato et al. [2018] broke three defenses for $\epsilon = 2$ non-targeted attacks on ImageNet [Guo et al., 2017, Liao et al., 2018, Xie et al., 2017], degrading their performance below 1%. Our free training algorithm is able to achieve 43% robustness against PGD attacks bounded by $\epsilon = 2$. Furthermore, we ran each experiment on a single workstation with four P100 GPUs. Even with this modest setup, training time for each ResNet-50 experiment is below 50 hours.

We summarize our results for various $\epsilon$'s and $m$'s in table 3 and fig. 4. To craft attacks, we used a step-size of 1 and the corresponding $\epsilon$ used during training. In all experiments, the training batch size was 256. Table 3 shows the robustness of Resnet-50 on ImageNet with $\epsilon = 2$. The validation accuracy for natural images decreases when we increase the minibatch replay $m$, just like it did for CIFAR in section 5.

The naturally trained model is vulnerable to PGD attacks (first row of table 3), while free training produces robust models that achieve over 40% accuracy vs PGD attacks ($m = 4, 6, 8$ in table 3). Attacking the models using PGD-100 does not result in a meaningful drop in accuracy compared to PGD-50. Therefore, we did not experiment with increasing the number of PGD iterations further.

Fig. 4 summarizes experimental results for robust models trained and tested under different perturbation bounds $\epsilon$. Each curve represents one training method (natural training or free training) with hyperparameter choice $m$. Each point on the curve represents the validation accuracy for an $\epsilon$-bounded robust model. These results are also provided as tables in the appendix. The proposed method consistently improves the robust accuracy under PGD attacks for $\epsilon = 2 - 7$, and $m = 4$ performs the best. It is difficult to train robust models when $\epsilon$ is large, which is consistent with previous studies showing that PGD-based adversarial training has limited robustness for ImageNet [Kannan et al., 2018].

Table 4: Validation accuracy and robustness of "free" and 2-PGD trained ResNet-50 models – both trained to resist $\ell_\infty$ $\epsilon = 4$ attacks. Note that **2-PGD training time is** $3.46\times$ **that of "free" training**.

| Model & Training | Evaluated Against | | | | Train time |
|---|---|---|---|---|---|
| | Natural Images | PGD-10 | PGD-50 | PGD-100 | (minutes) |
| RN50 – Free $m = 4$ | 60.206% | 32.768% | 31.878% | 31.816% | **3016** |
| RN50 – 2-PGD trained | **64.134%** | **37.172%** | **36.352%** | **36.316%** | 10,435 |

Table 5: Validation accuracy and robustness of free-$m = 4$ trained ResNets with various capacities.

| Architecture | Evaluated Against | | | |
|---|---|---|---|---|
| | Natural Images | PGD-10 | PGD-50 | PGD-100 |
| ResNet-50 | 60.206% | 32.768% | 31.878% | 31.816% |
| ResNet-101 | 63.340% | 35.388% | 34.402% | 34.328% |
| ResNet-152 | **64.446%** | **36.992%** | **36.044%** | **35.994%** |

**Comparison with PGD-trained models**

We compare "free" training to a more costly method using 2-PGD adversarial examples $\epsilon = 4$. We run the conventional adversarial training algorithm and set $\epsilon_s = 2$, $\epsilon = 4$, and $K = 2$. All other hyper-parameters were identical to those used for training our "free" models. Note that in our experiments, we do not use any label-smoothing or other common tricks for improving robustness since we want to do a fair comparison between PGD training and our "free" training. These extra regularizations can likely improve results for both approaches.

We compare our "free trained" $m = 4$ ResNet-50 model and the 2-PGD trained ResNet-50 model in table 4. 2-PGD adversarial training takes roughly $3.4\times$ longer than "free training" and only achieves slightly better results ($\approx 4.5\%$). This gap is less than 0.5% if we free train a higher capacity model (*i.e.* ResNet-152, see below).

**Free training on models with more capacity**

It is believed that increased network capacity leads to greater robustness from adversarial training [Madry et al., 2017, Kurakin et al., 2016b]. We verify that this is the case by "free training" ResNet-101 and ResNet-152 with $\epsilon = 4$. The comparison between ResNet-152, ResNet-101, and ResNet-50 is summarized in table 5. Free training on ResNet-101 and ResNet-152 each take roughly $1.7\times$ and $2.4\times$ more time than ResNet-50 on the same machine, respectively. The higher capacity model enjoys a roughly 4% boost to accuracy and robustness.

# 8 Conclusions

Adversarial training is a well-studied method that boosts the robustness and interpretability of neural networks. While it remains one of the few effective ways to harden a network to attacks, few can afford to adopt it because of its high computation cost. We present a "free" version of adversarial training with cost nearly equal to natural training. Free training can be further combined with other defenses to produce robust models without a slowdown. We hope that this approach can put adversarial training within reach for organizations with modest compute resources.

**Acknowledgements:** Goldstein and his students were supported by DARPA GARD, DARPA QED for RML, DARPA L2M, and the YFA program. Additional support was provided by the AFOSR MURI program. Davis and his students were supported by the Office of the Director of National Intelligence (ODNI), and IARPA (2014-14071600012). Studer was supported by Xilinx, Inc. and the US NSF under grants ECCS-1408006, CCF-1535897, CCF-1652065, CNS-1717559, and ECCS-1824379. Taylor was supported by the Office of Naval Research (N0001418WX01582) and the Department of Defense High Performance Computing Modernization Program. The views and conclusions contained herein are those of the authors and should not be interpreted as necessarily representing the official policies or endorsements, either expressed or implied, of the ODNI, IARPA, or the U.S. Government. The U.S. Government is authorized to reproduce and distribute reprints for Governmental purposes notwithstanding any copyright annotation thereon.

## Footnotes

[1]Xie et al. [2019] use 128 V100s and Kannan et al. [2018] use 53 P100s for targeted adv training ImageNet.

[2]Adversarial Training for Free code for CIFAR-10 in Tensorflow can be found here: `https://github.com/ashafahi/free_adv_train/`

[3]ImageNet Adversarial Training for Free code in Pytorch can be found here: `https://github.com/mahyarnajibi/FreeAdversarialTraining`

[4]Results based on the "adv_trained" model in Madry's CIFAR-10 challenge repo.

[5]Training using a non-iterative attack such as FGSM only doubles the training cost.

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
