[Supplementary Material · NeurIPS19_FreeAdv_Appendix.pdf]

# Adversarial Training for Free!

## A    Complete ImageNet results

Here we provide the results used for generating the ImageNet figures in tables 1 ~5.

Table 1: Val. accuracy and robustness of Resnet-50 models trained with $\epsilon = 3$.

| Training | Evaluated Against | | | |
|---|---|---|---|---|
|  | Natural Images | PGD-10 | PGD-50 | PGD-100 |
| Natural | 76.038% | 0.078% | 0.024% | 0.014% |
| Free $m = 2$ | 69.634% | 29.652% | 28.094% | 27.952% |
| Free $m = 4$ | 61.968% | 37.332% | 37.108% | 37.096% |
| Free $m = 6$ | 58.096% | 35.388% | 35.172% | 35.202% |
| Free $m = 8$ | 55.938% | 33.150% | 32.922% | 32.906% |

Table 2: Validation accuracy and robustness of Resnet-50 models trained with $\epsilon = 4$.

| Training | Evaluated Against | | | |
|---|---|---|---|---|
|  | Natural Images | PGD-10 | PGD-50 | PGD-100 |
| Natural | 76.038% | 0.072% | 0.014% | 0.010% |
| Free $m = 2$ | 68.126% | 23.902% | 21.224% | 20.978% |
| Free $m = 4$ | 60.206% | 32.768% | 31.878% | 31.816% |
| Free $m = 6$ | 55.988% | 30.804% | 30.282% | 30.250% |
| Free $m = 8$ | 52.190% | 29.004% | 28.624% | 28.608% |

Table 3: Validation accuracy and robustness of Resnet-50 models trained with $\epsilon = 5$.

| Training | Evaluated Against | | | |
|---|---|---|---|---|
|  | Natural Images | PGD-10 | PGD-50 | PGD-100 |
| Natural | 76.038% | 0.058% | 0.012% | 0.006% |
| Free $m = 2$ | 67.536% | 20.810% | 16.652% | 16.240% |
| Free $m = 4$ | 59.052% | 28.000% | 26.342% | 26.262% |
| Free $m = 6$ | 53.326% | 26.746% | 25.670% | 25.670% |
| Free $m = 8$ | 50.570% | 25.854% | 25.086% | 25.080% |

## B    The effect of batch-size

Our free training algorithm produces state-of-the-art results on CIFAR-10 and CIFAR-100 and results in robust models on ImageNet. We see that the ImageNet results are more sensitive to the replay

Table 4: Validation accuracy and robustness of Resnet-50 models trained with $\epsilon = 6$.

| Training | Evaluated Against | | | |
|---|---|---|---|---|
| | Natural Images | PGD-10 | PGD-50 | PGD-100 |
| Natural | 76.038% | 0.052% | 0.010% | 0.008% |
| Free $m = 2$ | 63.628% | 14.216% | 9.038% | 8.612% |
| Free $m = 4$ | 56.808% | 24.912% | 21.728% | 21.506% |
| Free $m = 6$ | 49.972% | 23.874% | 21.872% | 21.828% |
| Free $m = 8$ | 47.882% | 23.122% | 21.266% | 21.228% |

Table 5: Validation accuracy and robustness of Resnet-50 models trained with $\epsilon = 7$.

| Training | Evaluated Against | | | |
|---|---|---|---|---|
| | Natural Images | PGD-10 | PGD-50 | PGD-100 |
| Natural | 76.038% | 0.046% | 0.012% | 0.006% |
| Free $m = 2$ | 64.256% | 0.084% | 0.028% | 0.018% |
| Free $m = 4$ | 53.824% | 22.168% | 16.654% | 16.297% |
| Free $m = 6$ | 47.388% | 13.232% | 7.508% | 6.576% |
| Free $m = 8$ | 44.314% | 13.954% | 9.390% | 8.828% |

parameter $m$. While, our best results for CIFARs were with $m = 8$, our best ImageNet result is with $m = 4$. We believe that can be due to the ratio of number of classes ($N_c$) over batch-size ($b$). Our batch-size in the CIFAR experiments was 128. Since, we ran our ImageNet experiments on a single node with four GPUs, we were only able to use a batch-size of 256. If $N_c/b$ is large and $m$ is large, the probability that we do not see an example for some random class for more than $m$ iterations becomes large. This can result in catastrophically forgetting that class. To see the effect of batch-size in practice, we experimented with changing $b$ for CIFAR-100 and $m = 8$. In these experiments, we adjusted the learning-rate when we changed the batch-size. We used the linear learning-rate adjustment rule. The results which are consistent with our guess are summarized in fig. 1.

Figure 1: If the number of classes ($N_c$) is large, having a larger batch-size ($b$) can result in better robustness and generalization specially for larger values of $m$. In this experiment, we use $m = 8$ which yields the best result for CIFAR-100 ($N_c = 100$), and vary $b \in 16, 32, 64, 128$.

## C    Conventional K-PGD adversarial training

For completeness, we summarize the K-PGD $\ell_\infty$ algorithm in alg. 1.

---

**Algorithm 1** Standard Adversarial Training (K-PGD)

---

**Require:** Training samples $X$, perturbation bound $\epsilon$, step size $\epsilon_s$, maximization iterations per minimization step $K$, and minimization learning rate $\tau$

1: Initialize $\theta$
2: **for** epoch $= 1 \ldots N_{ep}$ **do**
3:     **for** minibatch $B \subset X$ **do**
4:         Build $x_{adv}$ for $x \in B$ with PGD:
5:             Assign a random perturbation
6:                 $r \leftarrow U(-\epsilon, \epsilon)$
7:                 $x_{adv} \leftarrow x + r$
8:             **for** $k = 1 \ldots K$ **do**
9:                 $g_{adv} \leftarrow \nabla_x l(x_{adv}, y, \theta)$
10:                 $x_{adv} \leftarrow x_{adv} + \epsilon_s \cdot \mathrm{sign}(g_{adv})$
11:                 $x_{adv} \leftarrow \mathrm{clip}(x_{adv}, x - \epsilon, x + \epsilon)$
12:             **end for**
13:         Update $\theta$ with stochastic gradient descent:
14:             $g_\theta \leftarrow \mathbb{E}_{(x,y) \in B}[\nabla_\theta \, l(x_{adv}, y, \theta)]$
15:             $\theta \leftarrow \theta - \tau g_\theta$
16:     **end for**
17: **end for**

---