[Reviews · NeurIPS 2019]

Reviewer 1



This paper proposes a new, more efficient method for performing adversarial training. The performance of the proposed training protocol is comparable to state-of-the-art results in adversarial training, while being efficient enough to adversarially train a model on ImageNet on a workstation. Experimental results are presented on CIFAR-10, CIFAR-100 and ImageNet. Originality: The idea of using the backward pass necessary for training to also compute adversarial samples seems indeed novel. Projected gradient descent (PGD) adversarial samples require multiple backward passes. In order to obtain strong adversarial samples for training, the same minibatch is used for training the model consecutively and to produce the PGD iterations each time on the updated gradient. The total number of epochs used for training is divided by the number of iterations on the same minibatch to ensure an equivalent number of training iterations as with natural training. Thus, the computation time for the proposed protocol is in the end comparable with that of natural training. Quality: The idea of "warm starting" each new minibatch with the perturbation values from the previous minibatch is not particularly founded, and no justification or ablation study is provided to analyze the impact of this choice. What happens when no warm start is used? How much does the final attack differ from the initialization value? Is this pushing the attack towards some notion of universal perturbation? The paper puts a strong accent on the fact that the proposed protocol is designed for untargeted adversarial training. It would be good to see a comparison with previous (targeted) results on ImageNet from [Xie et al., 2019] and [Kannan et al., 2018]. Some aspects of the experimental section are not fully convincing, as the attacks used for evaluation are arguably not too strong. The attacks against ImageNet (Table 3) seem to use $\epsilon=2$ (over 255?), which is too small a value to reflect the robustness of the model. Moreover, I was not able to find the exact parameters used when testing against the C&W attack (Table 1). Moreover, this attack was only evaluated on CIFAR-10. In most cases, evaluation against the PGD attack does not seem to use random restarts (except for one configuration int Table 1). This feature is known to make the attack considerably stronger. The paper mentions the SPSA black-box attack in the experimental section, but then fails to compare against it, claiming that it would not perform great anyway. The number of repetitions of the same minibatch $m$ seems to have a strong impact on both clean and adversarial accuracies (trade-off). How would one tune it efficiently in practice? Clarity: The paper is overall well written. Using both K-PGD and PGD-K notations can be a source of confusion. Significance: Provided that the method proposed in the paper is sound and obtains the claimed performance, it would offer a more efficient alternative to train a robust model. Minor remarks: - Lines 74-75: The main difference between BIM and PGD is actually the projection step performed by PGD (and giving the name of the method) but not by BIM. - Line 164: Possibly incorrect reference to Section 4. - Line 193: Extra word "reference". - Alg. 1, line 8: Extra square bracket. - Alg. 1, line 12: This should probably be a projection operation, not clipping, in order to generalize beyong $L_{\inf}$. [UPDATE] I would like to thank the authors for their detailed explanations and additional experiments. These have provided some additional clarity and should be included in the paper. In view of the rebuttal, some concerns still remain. I believe that testing the proposed adversarial training strategy against stronger attacks (e.g., using high confidence in C&W attack, larger eps budget for the others) would prove the robustness of the obtained model beyond a doubt. I am however increasing my rating from 5 to 7 in view of the rebuttal.

Reviewer 2



Originality: The paper has mainly one original idea - using the backward pass of backprop algorithm to also compute the adversarial example. On one hand, it is really impactful because the authors show empirically that it speeds up the training process while maintaining equal robustness to adversarial attacks, but on the other hand the idea itself isn't really outstanding. Quality: The paper gives experimental verification of the idea, and claim to achieve the state of the art robustness on CIFAR datasets. The paper also gives detailed results of the experiment like the training time taken, and show that it is indeed close the time taken for natural training. They also have a section explaining how the loss surface for their technique is flat and smooth; the adversarial examples for their technique look like the actual target class. These properties are also seen in standard adversarial training. Thus their technique is similar to the standard adversarial training even in these aspects. Therefore, quality of the paper is good. Clarity: The paper is well written. Significance: The significance would be really high because training robust models would be almost as fast as training non-robust models. This would greatly benefit the robust machine learning research. Having said that, other than this one idea, there aren't any other ideas or contributions of the paper.

Reviewer 3



I really enjoyed this paper. The idea of just simultaneously computing the gradient with respect to model parameters and the input images at the same time is simple and. elegant. It is surprising it works so well. I am also glad now that like the same PGD based technique works well on Cifar and ImageNet. Few Comments: 1) Since your method is so much faster it would be *great* to have error bars on your main tables/results. I am sure the variance is small but it would be nice to have that in the paper, it is also good practice for the community. 2) I am not sure what figure 3 is supposed to sho, it might be nice to contrast this with a non-robust model to really show the difference between the landscape between the two "robust"models (free and pgd) 3) You say there is *no* cost to increasing m in the cifar-100 section, but this is only true as long as m is less than the total number of epochs. I presume these algorithms wouldn't converge well in a low number of epochs (lets say 1 epoch). In fact it would be good to have that plot/table. 4) It would be nice to see the code to see how the fast gradient computation worked :) 5) Again since your method is so much faster it would be excellent to see how much more robustness you get on ImageNet by training adversarial on the *entire* Imagenet-11k (10 million example) training set. Recent work "Unlabeled Data Improves Adversarial Robustness" By Carmon et al has shown that adding more unlabeled data can improve robustness and "Adversarially Robust Generalization Requires More Data" by Schmidt et al postulates that robust generalization requires more data, so adversarial training on all 10 million ImageNet images which have roughly the same label quality as the standard 1.2 million training set might greatly improve performance. 6) It would be good to try the "stronger" attacks presented in https://github.com/MadryLab/cifar10_challenge just to see if these break your model in anyway 7) A plot like figure 7 in https://arxiv.org/pdf/1901.10513.pdf would be great, just another test to make sure your models are truly robust.

[Author Response · NeurIPS 2019]

We would like to thank reviewers **R1** , **R2** , and **R3** for their insightful comments. All will definitely help improving the
quality of the paper. Below, we tried to address as many comments as possible given the space and time constraints.

**R1** : <*Why "warm start" with the previous perturbation?*> Note that during the first replay step, with warm-start, we are
computing the gradient at a point corresponding to the original image plus a partially-random perturbation. In contrast,
without warm-start, the gradients are computed with respect to the clean image. For simplicity, consider $m = 2$ (*a.k.a.*
Free-2) which is similar to FGSM (the approximate gradient is only computed and applied once before moving on to the
next mini-batch). If we reset the perturbations to zero, the model doesn't become robust due to gradient masking (3.72%
accuracy against a PGD attack on CIFAR-10) and a simple rand+FGSM attack can break it. We experimented with
re-initializing the perturbation to be a random perturbation before the first replay step instead of re-using the perturbation
from the previous mini-batch and got similar robustness to re-using the perturbation from the previous mini-batch.
<$\epsilon = 2/255$ *used in ImageNet experiments is small.*> The results for other values (up to $\epsilon = 7$) are presented in Figure
4 and appendix. <*Exact parameters used when testing against the C&W attack (Table 1).*> We tested various values for
the CW attack parameters based on the Madry's implementation: $\text{maximize} - RELU(z_y(x_i + \delta) - z_w(x_i + \delta) + S)$,
where $z_y$ and $z_w$ are the correct and the largest incorrect logit, respectively. We use step-size=2, and $S = 0$ in the paper.
<*Why CW and random restart only evaluated on CIFAR-10?*> Since CIFAR-10 is one of the main benchmark problems
right now, we wanted to make sure of reporting all well-known attacks on this dataset. However per your request, we
have run an expanded set of experiments for the rebuttal, including CIFAR-100: 7-PGD trained (CW-100: 23.0%/
10-restart PGD-20: 22.6%) Free-8 (CW-100: 24.4%/ 10-restart PGD-20: 25.7%) and ImageNet: Free-8 (10-restart
PGD-20 40.0%, CW: 38.6%). <*Paper mentions SPSA, but fails to compare, claiming it would not perform great.*> We
started SPSA attacks, but results are still pending as SPSA is slow (over 24 hours/experiment). In preliminary results,
PGD was consistently stronger for all scenarios observed (this was also observed in the original SPSA paper, which
acknowledges the superiority of PGD when gradients are available to the attacker). <*How would one tune m?*> One
advantage of our method is having only a single hyper-parameter. It is fairly simple to tune $m$ using a coarse grid search
and training for a few epochs starting from a pre-trained model. <*Make source code/models publicly available.*> Our
official implementations based on both PyTorch and Tensorflow libraries are publicly available online. Moreover, our
results are replicated by 2 independent unofficial implementations. To maintain anonymity, we cannot provide links
here that reveal our Github IDs.

**R2** : <*It is really impactful because the authors show empirically that it speeds up the training process while main-*
*taining equal robustness...on the other hand, the idea itself isn't really outstanding.*> We feel that the simplicity of
implementation is a key attribute that makes our method useful. To add depth to a paper about a simple idea, we
tried our best to be thorough in our investigation. We observed the impact of minibatch replay on clean-trained nets,
studied the generative properties of free trained models, and validated the absence of gradient masking using landscape
visualizations. <*In the future, compare to "YOPO".*> YOPO appeared on arXiv after this paper was drafted, and
used different datasets and wider networks (WRN-34-10) with larger batch-sizes (256) than we do. As shown in our
supplementary, both of these factors increase robustness, so directly comparing to their arXiv results is difficult. To do a
direct comparison, we used our "Free" code (available on GitHub) to train WRN-34-10 using $m = 10$, batch-size=256.
We match their best reported result (48.03% against PGD-20 attacks for "Free" training *v.s.* 47.98% for YOPO 5-3).

**R3** : <**great** *to have error bars on your main tables/results.*> We have started to implement your great suggestion. For
$m = 8$ CIFAR-10, the natural accuracy is 85.95±0.14 and robustness against a 20-random restart PGD-20 attack is
46.49±0.19. Producing the error bars takes some time; we will try to have all error bars by the camera-ready deadline.
<*Figure 3 might be nice to contrast this with a non-robust model.*> The landscape of Non-robust models are not as
smooth and also have large peaks where the classification loss is relatively huge (Fig. 1). <*Code?*> It is publicly
available. See response to **R1** above. <*training adversarial on the* **entire** *Imagenet-11k*> Great suggestion! We will
try to add that to the camera-ready version. <*It would be nice to have some intuition or some justification for* **why**
*such a technique works so well.*> Our "free" updates are a coarse approximation of true adversarial training on the first
replay step, but they behave more and more like the true adversarial training updates after a few replay steps. In Fig.2,
we show that the gradients produced by "free" and "true" adversarial training become almost identical after 3 replay
steps. This study will appear in the supplementary.

Figure 1: loss landscape of nat trained.

Figure 2: We compute $\text{avg}(\text{sign}(\nabla_x l) == \text{sign}(\hat{\nabla}_x l))$ where $\hat{\nabla}_x l$ is the grad carried over from the previous replay step, and $\nabla_x l$ is from the current step. The updates become almost identical after 3 replay steps.

[Meta-Review · NeurIPS 2019]

The paper proposes using the backward pass of backprop algorithm to compute the adversarial example. This leads to computational time saving because instead of two backward passes. Overall, a good paper. The rebuttal was generally satisfying. It's encouraged to further consider the comments in the final version.